# Introducing Fluorescence-Guided Surgery for Pediatric Ewing, Osteo-, and Rhabdomyosarcomas: A Literature Review

**DOI:** 10.3390/biomedicines9101388

**Published:** 2021-10-04

**Authors:** Zeger Rijs, Bernadette Jeremiasse, Naweed Shifai, Hans Gelderblom, Cornelis F. M. Sier, Alexander L. Vahrmeijer, Fijs W. B. van Leeuwen, Alida F. W. van der Steeg, Michiel A. J. van de Sande

**Affiliations:** 1Department of Orthopedic Surgery, Leiden University Medical Center, Albinusdreef 2, 2333 ZA Leiden, The Netherlands; A.N.Shifai@lumc.nl (N.S.); M.A.J.van_de_Sande@lumc.nl (M.A.J.v.d.S.); 2Department of Surgery, Princess Maxima Center for Pediatric Oncology, Heidelberglaan 25, 3584 CS Utrecht, The Netherlands; B.Jeremiasse-4@prinsesmaximacentrum.nl (B.J.); a.f.w.vandersteeg@prinsesmaximacentrum.nl (A.F.W.v.d.S.); 3Department of Medical Oncology, Leiden University Medical Center, Albinusdreef 2, 2333 ZA Leiden, The Netherlands; A.J.Gelderblom@lumc.nl; 4Department of Surgery, Leiden University Medical Center, Albinusdreef 2, 2333 ZA Leiden, The Netherlands; C.F.M.Sier@lumc.nl (C.F.M.S.); A.L.Vahrmeijer@lumc.nl (A.L.V.); 5Percuros BV, 2333 CL Leiden, The Netherlands; 6Interventional Molecular Imaging Laboratory, Department of Radiology, Leiden University Medical Center, Albinusdreef 2, 2333 ZA Leiden, The Netherlands; F.W.B.van_Leeuwen@lumc.nl

**Keywords:** fluorescence-guided surgery, osteosarcoma, Ewing sarcoma, rhabdomyosarcoma

## Abstract

Sarcomas are a rare heterogeneous group of malignant neoplasms of mesenchymal origin which represent approximately 13% of all cancers in pediatric patients. The most prevalent pediatric bone sarcomas are osteosarcoma (OS) and Ewing sarcoma (ES). Rhabdomyosarcoma (RMS) is the most frequently occurring pediatric soft tissue sarcoma. The median age of OS and ES is approximately 17 years, so this disease is also commonly seen in adults while non-pleiomorphic RMS is rare in the adult population. The mainstay of all treatment regimens is multimodal treatment containing chemotherapy, surgical resection, and sometimes (neo)adjuvant radiotherapy. A clear resection margin improves both local control and overall survival and should be the goal during surgery with a curative intent. Real-time intraoperative fluorescence-guided imaging could facilitate complete resections by visualizing tumor tissue during surgery. This review evaluates whether non-targeted and targeted fluorescence-guided surgery (FGS) could be beneficial for pediatric OS, ES, and RMS patients. Necessities for clinical implementation, current literature, and the positive as well as negative aspects of non-targeted FGS using the NIR dye Indocyanine Green (ICG) were evaluated. In addition, we provide an overview of targets that could potentially be used for FGS in OS, ES, and RMS. Then, due to the time- and cost-efficient translational perspective, we elaborate on the use of antibody-based tracers as well as their disadvantages and alternatives. Finally, we conclude with recommendations for the experiments needed before FGS can be implemented for pediatric OS, ES, and RMS patients.

## 1. Introduction

Sarcomas are a rare heterogeneous group of malignant neoplasms of mesenchymal origin representing approximately 13% of all cancers in pediatric patients [1,2]. Sarcomas are generally subdivided into bone sarcomas and soft tissue sarcomas (STS) [3]. The most prevalent pediatric bone sarcoma is osteosarcoma (OS), with an annual incidence of 8–11 cases per million at 15–19 years of age [4], followed by Ewing sarcoma (ES), with an annual incidence of 9–10 cases per million at 10–19 years of age [5]. Rhabdomyosarcoma (RMS) is the most frequently occurring STS in the pediatric population, representing approximately 40% of all STS with an annual incidence of five cases per million below the age of 20 [6].

OS, ES, and RMS are commonly treated with multimodality therapy comprising surgery and (neo)adjuvant chemotherapy with or without radiotherapy [7,8,9,10,11]. For surgery, the current standard has been moved from amputations (with radical or wide margins) towards limb-salvage surgery with free margins [12,13]. Hence, the accuracy of surgical resection is an important prognostic factor for local recurrence-free and overall survival rates [11,14,15]. Although preoperative radiological imaging aids surgical planning, intra-operative margin assessment can be challenging, particularly when tumor tissue is surrounded by vital neurovascular structures or when tumors are located within deeper and more complex anatomical sites such as the pelvis or the head and neck region. Unfortunately, inadequate or positive resection margins are described in 10–40% of OS cases, 15–30% of ES cases, and in 20–30% of RMS cases [12,16,17,18,19,20]. Differences in local recurrence rates, 5-year overall survival, or 5-year event-free survival between adequate (defined as radical or wide) and inadequate (defined as marginal or intralesional) resection margins range from 20 to 25% in favor of adequate resection margins [11,12,15,16]. Apart from increasing local recurrence-free and overall survival rates, complete resections help reduce total dosages of adjuvant chemo- and or radiotherapy [11,17]. This is particularly relevant for pediatric patients as survivors face risks of common cancer treatment-related side effects, such as impaired growth and development, organ dysfunction, and secondary malignancies [21,22].

The increased local recurrence and decreased survival rate on the one hand and the increased risk of treatment-related side effects on the other hand indicate the necessity for adequate surgical resections. The real-time intraoperative visualization of malignancies could improve resection accuracy by aiding the surgeon discriminate between healthy and malignant tissue. Fluorescence-guided surgery (FGS) is one of the promising technological advances facilitating the visualization of tumors in real-time during surgery [23,24].

FGS exploits the advantages of near-infrared-I (NIR-I) light (750–1000 nm) or NIR-II light (1000–1700 nm), which have a tissue penetration of several millimeters to a centimeter deep [25]. Another advantage of NIR light is that almost no autofluorescence is exhibited in the NIR spectrum by biological tissue, which maximizes the potential tumor-to-background ratio of fluorescence when visualizing tumors [26,27]. In addition, the surgical field is generally not altered by NIR light, as the human eye is insensitive to NIR wavelengths [28]. The two main requirements for FGS comprise a fluorescent tracer and a dedicated camera system which captures light emitted by the tracer upon excitation with an appropriate light source [26]. FGS camera systems are manufactured by several companies; systems for open-, endoscopic- and/or robotic surgery were developed and are currently available [29].

Depending on which fluorescent tracers are applied, both non-targeted and targeted FGS is possible [28,30]. Indocyanine green (ICG) is the most used and investigated fluorescent dye for non-targeted FGS. Its benefits have been shown, amongst others, in assessing perfusion, identifying liver metastases and visualizing sentinel lymph nodes [23,31]. Targeted tracers contain fluorophores conjugated to cancer-specific targeting moieties such as antibodies, peptides or small molecule inhibitors [32,33]. While FGS has been investigated with promising results in various types of malignancies, information regarding its application in pediatric sarcomas such as OS, ES, and RMS is relatively scarce [34]. This review evaluates whether non-targeted and targeted FGS strategies hold promise for OS, ES, and RMS surgery. Necessities for clinical implementation, current literature, and the disadvantages of non-targeted FGS using ICG versus targeted FGS are evaluated. In addition, we provide an overview of tumor receptors that could be targeted in OS, ES, and RMS. Then, due to the time- and cost-efficient translational perspective, we elaborate on the use of antibody-based tracers as well as their disadvantages and alternatives. Lastly, we conclude with recommendations for the experiments needed before FGS can be implemented for pediatric OS, ES, and RMS patients.

## 2. Non-Targeted Fluorescence-Guided Surgery for OS, ES, and RMS Using Indocyanine Green

The indocyanine fluorescent dye ICG is already implemented for FGS in clinics. Currently, ICG is registered under two names: ICG-GREEN (Food and Drug Administration; FDA, Washington, DC, USA) and Verdye (European Medicines Agency; EMA, Amsterdam, The Netherlands. It can be administered with a maximum intravenous dose of 1.25 mg/kg for children aged 0–2 years, 2.5 mg/kg for children aged 2–11 years, and 5 mg/kg for children older than 11 years [35]. Once ICG is administered, it binds to plasma proteins, thereby increasing its hydrodynamic diameter to approximately 10 nm [36]. These complexes accumulate in tumors due to their leaky vascular capillaries, referred to as the enhanced permeability and retention (EPR) effect [37]. Once in the tumor, these molecules remain there due to their general characteristics such as size, shape, charge, and polarity, rather than tumor cell-specific binding.

ICG has been shown to be safe and accurate for the intra-operative visual identification of several tumor types in adults, such as colorectal liver metastasis, hepatocellular carcinoma, and brain tumors [27].

Although not applied for sarcoma resections, there is experience with ICG-guided surgery for pediatric patients [38]. Esposito et al. reported their results in 76 laparoscopic and/or robotic procedures (40 left varicocelectomies, 13 renal procedures, 12 cholecystectomies, 5 tumor excisions, 3 lymphoma excisions, 3 thoracoscopic procedures, 2 lobectomies, and 1 lymph node biopsy). They concluded that ICG-guidance is useful because it is easy to apply, safe, and allows for the better identification of anatomical structures as well as easier surgical dissection or resection in challenging cases. The technology is now also used in trial settings for pediatric surgical oncology [39].

### 2.1. Indocyanine Green for Sarcoma Resections

Only one study describes the use of ICG for various sarcoma resections in 26- to 79-year-old adults [40]. They included eleven patients, among which were one OS patient and one pleomorphic RMS patient who received 75 mg ICG 16–24 h before surgery. All sarcomas contained a fluorescent signal, except for the OS patient. However, this tumor was more than 90% necrotic due to neoadjuvant treatment. For the two patients, including the RMS patient, ICG fluorescence was of definite guidance, leading to extended tissue resection to improve the resection margin.

Multiple studies describe the use of ICG for the resection of pulmonary metastases, which also frequently occur in young sarcoma patients [41]. Predina et al. administered 5 mg/kg ICG 24 h preoperatively to 30 adult patients (23–79 years) suspected of pulmonary sarcoma metastases, including six OS patients, four ES patients, and two RMS patients [42]. They found that during thoracotomy or thoracoscopy, respectively, 88 and 89% of pulmonary sarcoma metastases showed fluorescence. Non-fluorescent (tumor-to-background ratio < 2) lesions were located deeper than 2 cm, corresponding with the maximum tissue penetration of light at this wavelength (<1 cm). Furthermore, ICG fluorescence identified additional occult lesions among which 88% were confirmed metastases and the others lymphoid aggregates. In addition, Keating et al. administered 5 mg/kg ICG 24 h preoperatively to eight adult patients (exact age not described) with the suspected pulmonary metastasis of various primary tumors including two OS patients [43]. Intraoperative thoracoscopic ICG fluorescence identified six of the eight preoperatively localized lesions. The missed nodules were the deepest from the pleural surface on the CT scan (1.8 cm and 1.6 cm). One additional nodule was identified by ICG fluorescence, which was a metastasis as confirmed by pathology. In addition, Okusanya et al. administered 5 mg/kg ICG 24 h preoperatively to 18 adult patients (29–79 years) with solitary pulmonary nodules that required resection [44]. Intraoperative thoracotomic ICG fluorescence identified 16/18 primary nodules with a maximum depth of 1.3 cm from the pleural surface. The two non-fluorescent nodules were identified by manual palpation and visual inspection. Additionally, ICG fluorescence also identified five additional subcentimeter nodules (minimum size 0.2 cm) of which two were metastatic sarcomas and three were metastatic adenocarcinomas.

Despite these results, it must still be assessed for which pediatric sarcoma types—often biologically different from sarcomas in adult patients—the application of non-targeted FGS using ICG could be useful [45]. St. Jude Children’s Research Hospital is currently performing a large phase 1 single-center trial for pediatric oncology patients, which will include 39 OS, 39 ES, and 39 RMS patients. The results of this trial (scheduled end-date December 2022) will represent a large step forwards in unraveling whether FGS using ICG could be of additive value for pediatric OS, ES, and RMS patients.

### 2.2. Pros and Cons of Fluorescence-Guided Surgery and Indocyanine Green for Patient and Surgeon

In general, FGS has several advantages when compared to other intra-operative detection methods. As mentioned in the introduction, it has a tissue penetration of several millimeters up to a centimeter, depending on the tissue type. It is relatively harmless compared with intraoperative computed tomography or radio-active agents. In addition, NIR-light emitted by NIR fluorophores is invisible to the naked eye and thus does not contaminate the surgical field nor does it leave long lasting tattoos, as is the case with blue dye [46]. Moreover, unlike the intraoperative histopathological examination of the surgical margin, FGS does not interrupt the surgical workflow [47]. Additional advantages have been reported for ICG specifically. ICG is relatively cheap and immediate reinjections are possible to assess perfusion when the fluorescent signal has diminished [48]. Furthermore, ICG is shown to be safe with only minor risks of adverse events, i.e., a risk of less than 1 in 10,000 of an anaphylactic reaction. Finally, ICG for FGS is generally given 24 h preoperatively, which is usually the moment patients are admitted to the hospital before undergoing tumor resection.

However, the general disadvantages of FGS include an additional investment for a dedicated camera system which may not be affordable for every hospital. In addition, bone tumors and nodules located deeper than 1 cm could still be missed due to the limited depth penetration of NIR fluorescence [25,49]. For the use of ICG, additional caveats and disadvantages have been described. First, there is not much scientific evidence regarding tumor-specific resections. Therefore, there is no proof that the use of ICG for tumor resections is beneficial for patient outcomes such as functional outcome, disease-specific local recurrence, and/or disease-specific survival. Secondly, since ICG is dissolved in a solution containing iodine, its application is contraindicated in patients with an iodine allergy or thyroid abnormalities, such as a clinical manifest hyperthyroidism or autonomous thyroid adenoma iodine [50]. In addition, patients with renal insufficiency might have an increased risk of anaphylactic reactions. Therefore, the advantages of ICG for patients with renal insufficiency (estimated GFR of <30 mL/min/1.73 m^2^) should be carefully weighed against the risk of potential adverse events. Additionally, for patients that would not be admitted 24 h preoperatively, the intravenous administration of ICG may be a burden from a logistical and financial point of view. Lastly, ICG fluorescence is associated with the EPR effect, which is known to be influenced by many factors, such as the tumor type, size, presence of necrosis, location, inflammation, and vascular mediators. Therefore, the signal intensity of ICG is unpredictable. False negativity could occur in cases with very small nodules, nodules with extensive necrosis or minimally viable tissue. In addition, false positivity could occur as well, for example in tissue with reactive changes or high levels of vascular permeability mediators such as bradykinin and prostaglandin [51,52].

## 3. Targeted Fluorescence-Guided Surgery for OS, ES, and RMS

Tumor-specific FGS does not depend on the tumor microenvironment, such as ICG with the EPR effect, but depends on tracers that bind to tumor-specific receptors. To select tumor-specific receptors that are appropriate for FGS, several characteristics have to be evaluated. The most important parameters for target selection are the following: targets should have been assessed in a large amount of tumor samples as this represents a measurement of evidence; a high percentage of tumor samples should actually express the tumor-specific target; when a tumor is positively stained, a high percentage of tumor cells should express the target; there should be a diffuse expression pattern of the tumor-specific target throughout the whole tumor and not in specific parts; the receptor should be preferably located on the cell surface of malignant cells to permit direct targeting with the possibility of internalization for a long-lasting signal; the tumor-specific receptor is still present after neoadjuvant therapy, which is important because neoadjuvant therapy is standard treatment for OS, ES, and non-pleiomorphic RMS; and the expression of the target should be absent or substantially less in adjacent normal tissue to adequately differentiate tumor from healthy tissue (Table 1).

### 3.1. Promising Tumor-Specific Fluorescent Agents for ES, OS, and RMS

Bosma et al. systematically reviewed 86 articles that studied 47 targets for FGS in primary ES tumors [53]. Cell surface protein expression was evaluated by Western blot or immunohistochemistry, and in descending order, the following nine targets were selected as the most promising for FGS: Cluster of differentiation 99 (CD99), C-X-C chemokine receptor type 4 (CXCR4), occludin, neuropeptide receptor Y1 (NPY1), LINGO-1, insulin like growth factor 1 receptor (IGF-1R), claudin-1, c-kit (also known as cluster of differentiation 117; CD117), and NOTCH receptor. Except for occludin, all previously mentioned targets have clinically available targeting moieties which in principle can be used for FGS in ES [53]. Still, further immunohistochemical studies that include both tumor and adjacent normal tissue should be performed to choose the most optimal candidate. In addition, more recent diagnostic markers, such as NKX2.2, could also be evaluated for their potential in FGS [54]. Nevertheless, the first steps were made to explore the promising targets for FGS in ES patients.

Systematic reviews selecting promising tumor-specific targets for OS and RMS have not been published to date. Therefore, we evaluated the literature to identify targets for FGS of OS and RMS. First, clinically available antibodies and their respective targeting antigens for these tumor types were identified from PubMed and clinicaltrials.gov (Appendix A). This search was restricted to therapeutic antibodies which have been previously or are currently evaluated in clinical trials because these antibodies can be relatively time- and cost-efficiently modified into fluorescent tracers [24,55]. Second, PubMed searches were performed to find important information for target selection (Appendix B). Here, we considered targets promising for FGS if the expression was evaluated in at least 20 tissue samples for a tumor subtype and more than 50% of the samples stained positive. When targets did not meet these two requirements, they were considered less promising. Although the remaining criteria in Table 1 are indeed important, solely data on sample size and the percentage of positive samples were available for each target. Therefore, only these two criteria could be assessed to determine the most promising targets.

Based on this strategy, the following seven targets were considered candidates for the FGS of OS: AXL receptor tyrosine kinase (AXL), B7 homolog 3 (B7-H3), cluster of differentiation 47 (CD47), disialoganglioside GD2 (GD2), transmembrane nonmetastatic melanoma protein B (gpNMB), IGF-1R, and vascular endothelial growth factor A (VEGF-A). Interestingly, all promising targets were demonstrated to internalize upon binding with an antibody (-derivative) in other tumor types, except for VEGF-A as it is not a cell-surface expressed receptor [56,57,58,59,60]. In contrast, three targets with clinically therapeutic antibodies were considered less promising for FGS. These were: human epidermal growth factor receptor 2 (HER2), programmed death-ligand 1 (PD-L1), and tumor endothelial marker 1 (TEM1) (Table 2).

An important nuance is that HER2, PD-L1, and VEGF-A were investigated in a large number of (pre)clinical studies. The remaining targets were evaluated considerably less. Publication bias might have had an impact on the published results concerning these targets.

For RMS, less literature is published regarding the expression of targets with clinically available antibodies. Based on the criteria in Table 1, three promising targets were selected: the cluster of differentiation 56 (CD56), IGF-1R, and VEGF-A (Table 3). Of these, IGF-1R has been demonstrated to internalize [57]. Interestingly, all studies are mainly investigated alveolar RMS and/or embryonal RMS. These are the subtypes which most frequently occur in pediatric RMS patient. In contrast, B7-H3 and TEM1 were considered less promising for FGS in RMS (Table 3).

Combining the results from the systematic review by Bosma et al. with Table 2 and Table 3, IGF-1R seems the only target that is simultaneously promising for OS, ES, and RMS [53]. This suggests that a fluorescent dye conjugated to a clinically available antibody targeting IGF-1R (Appendix A) could be applicable for the majority of pediatric OS, ES, and RMS sarcoma patients.

### 3.2. Obstacles Regarding the Selection of Tumor-Specific Targets for Fluorescence-Guided Surgery

Data presented in the previous paragraphs are based on immunohistochemical studies which have been performed for other purposes than FGS. Therefore, the interpretation of the results of those studies for FGS purposes should be done with caution. The comparison of immunohistochemical studies is complicated as large inconsistencies in reported target expression exist between various studies. This can be due to the application of different antibodies for the same target but against various epitopes, due to differences in staining protocols, or due to inter-/intra-tumoral heterogeneity [61,62,63]. Moreover, the use of immunohistochemical evaluation in formalin-fixed paraffin-embedded (FFPE) samples with the intention to target the same protein on living cells for imaging purposes has multiple disadvantages. A prominent cause of the loss of antigenicity is formalin fixation which generates crosslinks between adjacent proteins that result in the steric interference of antibody binding to the respective epitope [64]. Furthermore, bone tissue requires decalcification, which is known to impair antigen retrieval and alters the immunohistochemical staining intensity [65,66].

To overcome the aforementioned obstacles, studies with the aim of target evaluation for FGS should be performed. Here, the staining of undecalcified fresh frozen sections (FFS) could be advantageous [62,67,68]. Unfortunately, many clinical FFS specimens are not readily available in this form, which complicates acquiring large sample sizes; storage is expensive and requires an advanced infrastructure; and tissue morphology is less well preserved over time compared to FFPE [69,70,71].

### 3.3. Translational Perspective of Targeted Fluorescence-Guided Surgery for OS, ES, and RMS

The previous sections provide an overview of potential tumor-specific targets for OS, ES, and RMS. As fluorophores by themselves generally lack tumor specificity, they could be conjugated to a targeting moiety such as a monoclonal antibody, a peptide, or small molecule inhibitors [32]. Antibody-based tracers are most often investigated for FGS as monoclonal therapeutic antibodies against a wide variety of targets are readily available and can be repurposed for FGS by fluorophore conjugation [26]. For this purpose, various fluorophores are available, and some are clinically approved, paving the way for implementing FGS to optimize surgical oncology [29].

Several therapeutic antibodies binding to candidate targets for OS, ES, and RMS are available (Appendix A). Additional therapeutic antibodies that are currently being investigated in clinical trials are described in Appendix A. Because most therapeutic antibodies are human or have been humanized, they are reported with tolerable safety profiles. Moreover, it should be noted that doses of antibodies injected for FGS are substantially (approximately 10–100×) lower compared to therapeutic doses. Consequently, the serum concentration of the antibody (conjugated to a fluorophore) is lower when used for FGS and little or no toxicity is expected [72]. Moreover, some of the therapeutic antibodies evaluated for therapy in pediatric OS, ES, and RMS patients have already been repurposed for FGS and accurately visualized tumor tissue in other cancer types [73,74,75,76,77]. Although it is important to notice that HER2 is considered a less promising target in OS (Table 2), Trastuzumab-IRDye800CW targeting HER2 has imaged breast cancer and could be tested in OS patients as well [75]. More encouragingly, Bevacizumab-IRDye800CW targeting VEGF-A was successful for FGS in adult soft tissue sarcoma patients [77]. Due to the presence of VEGF-A in pediatric OS and RMS (Table 2 and Table 3), testing Bevacizumab-IRDye800CW is a relatively straightforward option which could pave the road towards the clinical implementation of FGS in pediatric OS and RMS patients.

**Table 2 biomedicines-09-01388-t002:** Characteristics of targets evaluated for fluorescence-guided surgery in osteosarcoma.

Targets	Tissue Samples	% Positive Samples	% Positive Cells	Expression Altered after Neo-Adjuvant Therapy	(Adjacent) Healthy Tissue	Internalization ^1^	References
Promising targets ^2^							
AXL	40 TMA (RM) + 6 TB	75–83%	N.D.	N.D.	TMA (RM): 20% of 40 weakly positive	Yes	[58,78,79]
B7-H3	61 PR	92%	Median: 90%	N.D.	61 cases matched healthy tissue negative staining	Yes	[60,80]
CD47	20 RM	85%	N.D.	N.D.	WB: substantially lower protein levels in 6 healthy tissues compared to tumor tissue	Yes	[57,81]
GD2	55 TMA (from TB, PR, RR)	95–100%	N.D.	No, target expression remains	N.D.	Yes	[59,82,83]
gpNMB	67 TMA (from TB, PR, RR)	93%	>66% positive tumor cells in 37% of samples	Probably not ^3^	N.D.	N.D.	[84]
IGF-1R	84 TS	86%	50–75% positive cells in 24% of samples	N.D.	WB: substantially lower protein levels in healthy tissue compared to tumor tissue	Yes	[56,85]
VEGF-A	466 TMA (TB + PR + RR) + TB + PR + RR	Average of 59.9% (range 15–96%)	>50% in 11–38.8% of samples	Uncertain, varying results in different studies	N.D.	No	[86,87,88,89,90,91,92,93,94,95,96,97,98]
Less promising targets ^2^							
HER2	1 systematic review: 934 TB + PR + RR	Average 42% (range) 13–60%	<50%	N.D.	10 healthy bone samples from fractures were negative	Yes	[87,99,100,101]
PD-L1	418 TMA (TS) + TB + PR + RR	Average of 32.5% (range 0–85%)	<25% ^4^	Yes, decreased ^5^	N.D.	Yes	[66,102,103,104,105,106,107,108,109,110]
TEM1	11 TS	63.6%	N.D.	N.D.	N.D.	Yes	[111,112]

Abbreviations: TMA—tissue microarray; RM—resection material (unspecified whether it was a primary tumor or a recurrence); TB—tissue biopsies; N.D.—not described; WB—Western blot; PR—primary resection material; RR—recurrence resection material; TS—tissue samples (unspecified whether it is tissue biopsy or resection sample). ^1^. Endocytosis of an extracellular molecule upon binding to a specific protein on the cell surface. While not described in OS, it was demonstrated that AXL, CD47, GD2, IGF-1R, and B7-H3 were internalized upon binding with an antibody (derivative) in other tumor types [56,57,58,59,60]. ^2^. Targets were considered promising for FGS, if the expression was evaluated in at least 20 tissue samples for a tumor subtype and more than 50% of the samples stained positive. When targets did not meet these two requirements, they were considered less promising. ^3^. The staining intensity was similar in tissues from biopsies and resection material. Since OS patients commonly receive neoadjuvant treatment, gpNMB expression is most likely not altered after neoadjuvant treatment. ^4^. One study described the percentage of positive tumor cells. 84 tissue slides were assessed, among which 12 tissue slides were positive. In these, less than 25% of cells stained positive for PD-L1 110. ^5^. One study compared diagnostic biopsies with the corresponding tissue from subsequent primary resections after neoadjuvant chemotherapy (methotrexate, cisplatin, and doxorubicin). While 53% of pre-neoadjuvant treatment biopsies stained positive for PD-L1, none of the resection samples did [65].

**Table 3 biomedicines-09-01388-t003:** Characteristics of targets evaluated for fluorescence-guided surgery in rhabdomyosarcoma.

Targets	Tissue Samples	% Positive Samples	% Positive Cells	Expression Altered after Neo-Adjuvant Therapy	(Adjacent) Healthy Tissue	Internalization ^1^	References
Promising targets ^2^							
CD56	117 TMA (TS) + TS	Average 96% (range 90–100%)	>50% positive cells in >75% of sample	N.D.	N.D.	N.D.	[113,114,115,116,117]
IGF-1R	124 TB + RM	63–83%	N.D.	N.D.	N.D.	Yes	[56,118,119]
VEGF-A	145 TMA (PR) + PR + RR	59–70%	N.D.	N.D.	N.D.	N.D.	[120,121]
Less promising targets ^2^							
B7-H3	4 patient-derived xenografts	100%	Average H-score 108 (range 49–150) ^3^	N.D.	N.D.	Yes	[60,122]
TEM1	126 TMA (TB)	31%	N.D.	N.D.	N.D.	Yes	[112,123]

Abbreviations: TMA—tissue microarray; TS—tissue samples (unspecified whether it was tissue biopsy or resection sample); N.D.—not described; TB—tissue biopsies; RM—resection material (unspecified whether it was a primary tumor or a recurrence; PR—primary resection material; RR—recurrency resection material. ^1^. Endocytosis of an extracellular molecule upon binding to a specific protein on the cell surface. While not described in RMS, it was demonstrated that CD56, TEM1, and B7-H3 internalized upon binding with an antibody (derivative) in a different tumor type [56]. ^2^. Targets were considered promising for FGS if the expression was evaluated in at least 20 tissue samples for a tumor subtype and more than 50% of the samples stained positive. When targets did not meet these two requirements, they were considered less promising. ^3^. The H-score ranged from 0 to 300 and is a semiquantitative score obtained by determining the staining intensity score (ranging from 0 to 3) of each tumor cell and applying the formula [1 × (% cells 1+) + 2 × (% cells 2+) + 3 × (% cells 3+)].

## 4. Disadvantages of Using Antibodies for Fluorescence-Guided Surgery and Their Alternatives

Although antibodies bind to their target with high affinity and specificity, the large size of antibodies (150 kDa) is expected to limit tumor penetration and establish a long blood half-life. Therefore, the use of antibodies as targeted tracers for tumor-specific FGS necessitates a turnover time of several days to obtain an optimal tumor-to-background ratio [124]. For patients, this implicates an extra preoperative hospital visit for tracer administration. Apart from the extra costs and logistic planning, this leads to an extra burden for pediatric OS, ES, and RMS patients. Therefore, smaller targeting moiety alternatives such as antibody fragments, peptide conjugates and small molecule conjugates may be advantageous for FGS [33]. Briefly, tracers smaller than 60 kDa are generally cleared via the kidneys instead of via the liver, resulting in faster blood clearance from non-targeted tissues. Consequently, a high tumor-to-background contrast can be attained within hours after the administration of a tumor-specific smaller targeting moiety alternative-based tracer [26,33,125].

As mentioned before, deeper located tumors could still be missed by these tracers [25,49]. To this end, hybrid radionuclear/fluorescence imaging combined into a single tracer is attractive. The nuclear component can be of added value for initial intra-operative navigation towards the tumor using a gamma-detecting probe. When the tumor is reached, fluorescence could provide high resolution visual guidance for precise resection [125]. The integration of both modalities onto a tracer can be effectuated in several ways. The conjugation of a fluorophore and chelator for radioactive labeling at different positions of the tracer is one option. Examples include colorectal carcinoma imaging using fluoresceinated monoclonal antibodies against carcinoembryonic antigen which were also labeled with 125I, and the conjugation of IRDye800CW and 111In to Girentuximab for the visualization of clear-cell renal-cell carcinomas [126,127]. Combining the NIR fluorophore and the radiolabel into a single structure is another option [128]. This would be advantageous since this is in most cases a site-specific labeling method which implies that the properties of the tracer cannot vary, creating a predictable tracer with regards to its biodistribution. In addition, this is often the only possibility when working with small peptides or chemical molecules, as multiple conjugation sites may not be available. An example of this technique is the hybrid nuclear and NIR fluorescent PSMA tracer in prostate cancer [129]. However, these single structures are more complex to design, as opposed to the more straightforward separate conjugation of the fluorophore and chelator to an antibody.

## 5. Future Perspective and Recommendations for Fluorescence-Guided Surgery in OS, ES, and RMS Patients

In general, the standard preclinical work preceding the application of FGS consists of the following: target selection, target evaluation, tracer development, in-vitro evaluation, in vivo evaluation with animal experiments, and clinical in-human experiments. Because FGS for pediatric OS, ES, and RMS patients is still in its infancy, this review reveals an overview of promising targets as a basic first step. A follow-up would be to evaluate the expression of these targets with immunohistochemistry in a comparative setting using intra-patient sets of tumor and control tissue. Ideally, fresh surgically resected- or frozen tissue with representative intra- and intertumoral heterogeneity should be used to determine the level and diffusivity of target expression [130,131]. A possible advantage of frozen tissue would be that the antigenicity of its targets may be comparable to those on in-human tumors. The disadvantage is probably the logistics of obtaining these samples in a (large) representative cohort which would involve prospective sampling in a multicenter setting. Therefore, we recommend first evaluating the expression of targets with immunohistochemistry using FFPE samples, given its availability and relatively low cost. Subsequently, the expression of promising targets should be evaluated in fresh surgically resected- or frozen tissue. This enables the systematic, cost-effective, and valid selection of candidate targets for FGS. Then, fluorescent tracers should be developed based on the most favorable target evaluated by immunohistochemistry. Clinically available antibodies conjugated to fluorophores that bind to these targets allow faster and cost-efficient translation to clinics. Afterwards, preclinical (mouse) models can be used to show in-vivo efficacy. However, the major disadvantages of genetically altered mice are the absence of human tumor microenvironment and the difficulty of mimicking intertumoral heterogeneity. Therefore, results cannot be directly extrapolated to the clinical setting. Ultimately, the assessment of the additive value of FGS can only be performed in humans.

## 6. Conclusions

We created an overview of the potential of non-targeted and targeted FGS for pediatric OS, ES, and RMS patients. Although ICG has been shown to be safe in pediatric patients, the results of an ongoing clinical trial (scheduled end-date of December 2022) will reveal whether FGS using ICG is accurate for the intra-operative visual identification of pediatric OS, ES, and RMS patients. As FGS using ICG is based on the EPR effect, false negative and false positive signals could occur. Therefore, targeted FGS might be a better alternative. We listed the promising tumor-specific targets for FGS in pediatric OS, ES, and RMS patients. The conjugation of fluorophores to clinically available antibodies that bind to these targets may result in safe and tumor-specific tracers that could improve tumor resection success rates. Evaluating the already clinically tested fluorescent-labeled antibody Bevacizumab-IRDye800CW in pediatric OS and RMS patients is a relatively straightforward option because this has already shown promising results in adult soft tissue sarcomas. However, the development of a fluorescent-labeled IGF-1R antibody would be ideal because this seems to be the only promising target for pediatric OS, ES, and RMS simultaneously. Due to disadvantages of antibody-based tracers, smaller targeting moiety alternatives for tracer development should be investigated. In addition, surgical navigation with the use of hybrid radionuclear/fluorescence tracers could be investigated to detect deeper located and hidden tumors. In conclusion, FGS has the potential to optimize OS, ES, and RMS treatment, but more research remains to be done before this promising technique can be implemented for OS, ES, and RMS patients.

## Figures and Tables

**Table 1 biomedicines-09-01388-t001:** Important parameters for target selection.

- Target expression is evaluated in a large amount of tumor samples as this represents a measurement of evidence - A high percentage of evaluated samples display positive staining - When a tumor is stained positively, a high percentage of tumor cells express the target - The target is expressed diffusely throughout the whole tumor - The target is located on the cell surface of malignant cells - Expression of the target persists after neoadjuvant therapy - Target is minimally or not expressed in adjacent healthy tissue

## Data Availability

Not applicable.

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
