# Peer review of "Introducing Fluorescence-Guided Surgery for Pediatric Ewing, Osteo-, and Rhabdomyosarcomas: A Literature Review"

_biomedicines, 2021, doi:10.3390/biomedicines9101388_

Round 1

Reviewer 1 Report

Dear Editor,

I read the article from Rijs and coll on the purpose of Fluorescence

 Introducing Fluorescence-Guided Surgery for Pediatric Ewing,  Osteo- and Rhabdomyosarcomas: A Literature Review.

The paper is  a complete analysis  of the biochemical and pharmacological basis of the direct and indirect fluorescine marking technique to improve the surgical results in sarcomas.

- The review of the Literature is wide and in depth analysis of the topics is complete

-Some new perspectives are presented to improve the results of surgical management of pediatric sarcomas

- The English form is excellent

On the other   my  fundamental question is: do we really need such a complex technique to improve the local control of OS, ES and RMS?

The reasons of my doubts  are the following:

  • Do the patients die from local relapse or from distant metastasis? Until now all the studies confirm that the cause of death in OS,ES and RMS are distant metastasis. We know that local relapse can sometimes prelude to metastatic disease , but the preoperative staging of sarcomas using  CT scan , MRI and sometimes PET scan allows a precise definition of the primary sarcoma margins. The old but always accepted marking  with ink of the surgical sample  is still  widely accepted to define the close margin areas . The costs of the two techniques are deeply different in favour of ink marking.
  • Does this technique is useful in sarcomas where, following  the great lesson from Enneking ,wide or  radical margins are always requested  to consider the intervention as adequate. Marginal and  intralesional resection  must be reoperated in any case. The fluorescine technique does really change the approach to these tumor?
  • Apart the name “sarcoma”, what have in common OS, ES and RMS?

 Each of these  three diseases  have different morphology, biology and prognosis.    The antigenic characteristics are very different, and in many cases , as the Authors said, they do not express a specific target  at all. How  can we conjugate fluorescine with a monoclonal antibody which could mark only a small number of cells even in the same tumor?

  • The very good and complete review of Literature does not present a sufficient number of published studies of positive application of fluorescine technique on sarcomas. The Authors report one experience in pulmonary metastasis and one unpublished  application on primary tumor.

( Reference 41). No data on RFS and OS are presented.

 On the basis of this weak  evidence it is difficult to accept the fluorescine marking  as a standard technique in pediatric sarcomas.

  • The international protocols of integrated therapy for OS, ES and RMS  provide for the use of neoadjuvant chemotherapy before surgery in order to cause the highest level of necrosis  into the tumor ( see Huvos for OS and Picci for ES).The highest is the necrosis the better is prognosis.

 After a good response to chemotherapy ( i.e. >90% necrosis) how can apply the fluorescine technique on a necrotic tissue?

In conclusion I totally agree with the Authors who at point 5  of the manuscript affirm:

“In general, the standard preclinical work up before applying FGS consists of the following: target selection, target evaluation, tracer development, in-vitro evaluation, in  vivo evaluation with animal experiments and clinical in-human experiments. Because  FGS for pediatric OS, ES, and RMS patients is still in its infancy, this review reveals an  overview of promising targets as a basic first step.”

This review  can be considered a  introducing approach to the fluorescine marking techique in sarcoma resection.

Author Response

Dear reviewer 1,

With regards to reference 41, the reference (Nicoli et al. 2021; PMID 32224746) was checked, and data on RMS and OS was presented. The first section of the results has been copied and pasted here:

Eleven patients (6 women and 5 men) with an age range of 26

to 79 years were studied. Anatomical location was in the lower limb

(n . 4), the upper limb (n . 4), groin (n . 1), chest wall (n . 1), and

pelvis (n . 1). The final tumor histology was pleomorphic sarcoma

(n . 1), myxofibrosarcoma (n . 5), chondrosarcoma (n . 1),

synovial sarcoma (n . 1), leiomyosarcoma (n . 1), pleomorphic

rhabdomyosarcoma (n . 1), and osteosarcoma (n . 1).

Unfortunately, the whole article could not be added as an attachment because only 1 file can be uploaded (which is the complete response to Reviewer 1).

Yours sincerely,

Zeger Rijs

Reviewer 2 Report

This is a very well written and comprehensive review of existing studies on Fluorescence for paediatric sarcomas. This offers the reader an overview of what has been done in the field and what is the prospective on the subject.

Author Response

Dear reviewer 2,

On behalf of my colleagues, I would like to thank you for reviewing our manuscript entitled “Introducing Fluorescence-Guided Surgery for Pediatric Ewing, Osteo- and Rhabdomyosarcomas: A Literature Review”. We did not attach a response to the reviewer's comments because there were no comments.

Yours sincerely,

Zeger Rijs

Round 2

Reviewer 1 Report

This article  is a complete  review of the literature on fluorescence-guided surgery for pediatric Ewing, Osteo- and Rhabdomyosarcomas.

Interesting suggestions but nothing definitive  as the Authors state in the conclusions.

The article is  not a great advance in the knowelwdge  of surgical management of soft tissue and bone sarcomas of the children. 

The status of the margins can be precisely  detected and described only at  the histological  analysis and doing a meticolous staging of the disease, before surgery

The present  review could  be useful to plan a randomized study  comparing the classical intervention versus fluorescine  guided  surgery, if the rarity of the 3 sarcomas  allows a randomization .

Author Response

Dear reviewer,   On behalf of my colleagues, I would like to thank the reviewer for the comments and agree that this review review could eventually be useful to plan a randomized study  comparing the classical intervention versus fluorescence-guided surgery, if the rarity of the 3 sarcomas  allows a randomization.   On behalf of the authors,

Yours sincerely,

Zeger Rijs